# Surgical Flow Masked Autoencoder for Event Recognition

**Mayar Lotfy Mostafa**[1,2]                                        MAYAR.MOSTAFA@TUM.DE
**Anna Alperovich**[1]
**Dmitrii Fedotov**[1]
**Ghazal Ghazaei**[1]
**Stefan Saur**[3]
**Azade Farshad**[2,4]
**Nassir Navab**[2,4]

[1] *Carl Zeiss AG, Corporate Research & Technology, Oberkochen, Germany*

[2] *CAMP, Technical University of Munich, Garching, Germany*

[3] *Carl Zeiss Meditec AG, Oberkochen, Germany*

[4] *Munich Center for Machine Learning (MCML), Munich, Germany*

**Editors:** Accepted for publication at MIDL 2025

## Abstract

Recognition and forecasting of surgical events from video sequences are crucial for advancing computer-assisted surgery. Surgical events are often characterized by specific tool-tissue interactions; for example, "bleeding damage" occurs when a tool unintentionally cuts a tissue, leading to blood flow. Despite progress in general event classification, recognizing and forecasting events in medical contexts remains challenging due to data scarcity and the complexity of these events. To address these challenges, we propose a method utilizing video masked autoencoders (VideoMAE) for surgical event recognition. This approach focuses the network on the most informative areas of the video while minimizing the need for extensive annotations. We introduce a novel mask sampling technique based on an estimated prior probability map derived from optical flow. We hypothesize that leveraging prior knowledge of tool-tissue interactions will enable the network to concentrate on the most relevant regions in the video. We propose two methods for estimating the prior probability map: (a) retaining areas with the fastest motion and (b) incorporating an additional encoding pathway for optical flow. Our extensive experiments on the public dataset CATARACTS and our in-house neurosurgical data demonstrate that optical flow-based masking consistently outperforms random masking strategies of VideoMAE in phase and event classification tasks. We find that an optical flow encoder enhances classification accuracy by directing the network's focus to the most relevant information, even in regions without rapid motion. Finally, we investigate sequential and multi-task training strategies to identify the best-performing model, which surpasses the current state-of-the-art by 5% on the CATARACTS dataset and 27% on our in-house neurosurgical data.

**Keywords:** Surgical Phase Recognition, Optical Flow, Masked Autoencoders, Adverse Events Recognition.

## 1. Introduction

Surgical workflow analysis provides valuable insights into the intricate sequence of events during surgical procedures. By understanding and analyzing these events, it is possible to enhance performance, optimize patient care, and improve training for medical professionals.

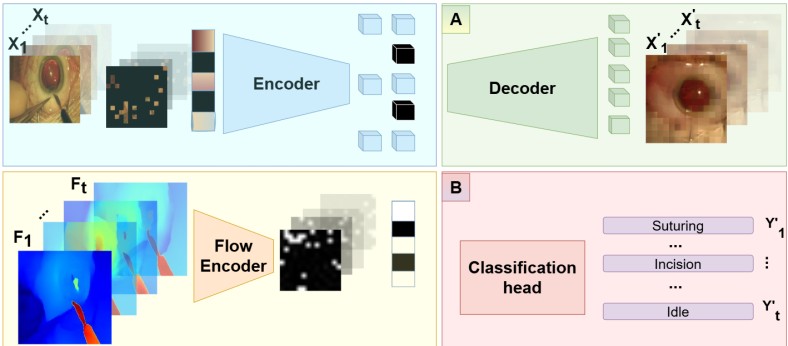

Figure 1: Overview of the Flow Masked Autoencoder architecture. The model receives input frames $X_T$ and their corresponding optical flow frames $F_T$. The bottom left section shows the learned optical flow encoding mask, which is applied to the input before feeding the masked image into the encoder. Path (A) denotes the decoder head for reconstructing the input $X_T$, while Path (B) denotes the classification head for predicting surgical phases $Y_T$.

It serves two main purposes: aiding intraoperative decision-making by recognizing the current surgical phase and guiding timely assistance, as well as enabling retrospective analysis for education, quality control, and workflow optimization. However, automated surgical workflow recognition is not widely adopted in operating rooms due to challenges related to robustness and reliability in complex surgical environments.

Operating room scenes are inherently intricate, often containing numerous irrelevant elements such as unused instruments, the surgeon's hands, and holders that obscure the main regions of interest. Prior studies on cataract surgery videos (Yu et al., 2019) have demonstrated that leveraging tool information significantly enhances phase segmentation performance. Similarly, DeepPhase (Zisimopoulos et al., 2018) highlights the importance of tool features in improving surgical workflow recognition. More recently, dynamic scene graphs have been employed to represent summaries of surgical scenes and specific tool-anatomy interactions for better surgical workflow recognition (Holm et al., 2023; Köksal et al., 2024). However, these methods rely on highly detailed annotations of surgical scenes, which require substantial resources. Building on these insights, we extend this approach by focusing not just on tool features but also on the critical tool-tissue interactions that define surgical workflows. These interactions capture the core dynamics of surgical procedures, making them invaluable for accurate workflow analysis. Video Masked Autoencoder (VideoMAE)-based solutions have shown that masking strategies can effectively identify relevant regions in video data by learning robust spatiotemporal representations (Tong et al., 2022). However, existing approaches are often generic and fail to address the unique challenges posed by surgical videos, where key interactions are often localized and obscured by irrelevant details.

In this work, we introduce SurgFlowMAE, a novel optical flow-guided masking strategy that leverages tool-tissue interaction dynamics to enhance VideoMAE for surgical workflow analysis. Our approach introduces a smart masking strategy that leverages optical

flow information to identify and focus on regions with influential motion, such as tool-tissue interactions while ignoring irrelevant areas. Specifically, we develop two strategies for incorporating optical flow. First, we use its normalized magnitude directly, to create a masking probability map. Second, we incorporate an additional encoding pathway for optical flow, allowing the model to learn the most relevant regions in the scene. Finally, with both approaches we use the estimated map as a prior probability for mask sampling in MAE. This ensures that the masked autoencoder focuses on extracting meaningful features, improving downstream performance on such tasks as phase segmentation and adverse event classification.

We evaluate SurgFlowMAE on two distinct surgical video datasets, highlighting its applicability across different surgical domains. SurgFlowMAE achieves state-of-the-art (SOTA) performance on the task of phase segmentation on the CATARACTS dataset, outperforming methods that incorporate comprehensive surgical scene information through complex graph-based representations (Köksal et al., 2024; Holm et al., 2023). We demonstrate the generalizability and flexibility of our approach through evaluations on distinct surgical datasets, achieving up to 5% improvement on the CATARACTS dataset and setting a new benchmark for adverse effect classification in Neurosurgery.

## 2. Related Work

Recent works in event prediction from video sequences focus on fusing spatiotemporal information into relevant features for classification. The Masked Autoencoder (Tong et al., 2022) is a popular choice for video understanding due to three main reasons. First, it reconstructs missing parts using contextual information, improving comprehension of complex events across frames. Second, its transformer architecture facilitates robust representation learning in the image domain. Lastly, MAE enables unsupervised pre-training without requiring labels, reducing annotation efforts crucial for video data.

**Event recognition in the wild** (Mao et al., 2023) proposed a motion-aware masking strategy (MAMP) for 3D human action recognition that predicts masked joint motion from spatiotemporal video sequences, adding semantic information to the masking process. (Sun et al., 2023) focuses on learning video representation by reconstructing the motion of masked regions, aiming to recover motion trajectories instead of appearance, using the semantics of masked objects inferred from visible patches. (Bandara et al., 2023) adapted the REIN-FORCE algorithm (Williams, 1992) to sample visible tokens from a categorical distribution. Their proposed network maximizes expected reconstruction error through policy gradients, surpassing fixed distribution methods. (Huang et al., 2023) introduces a motion-guided masking strategy using optical flow for consistent masking volume. While their approach offers an online solution for flow masking, it is slower than traditional VideoMAE (Tong et al., 2022).

**Surgical Workflow Recognition** The early EndoNet by (Twinanda et al., 2016) offers a method for surgical phase classification and tool position detection in a multi-task framework; and it outperforms single-task methods on the Cholec80 dataset. The authors show that incorporating the tool presence task enhances EndoNet's ability to learn more discriminative features. SV-RCNet (Jin et al., 2017) combines a CNN and RNN for the

Cholec80 dataset. Based on integrating ResNet (He et al., 2016) and LSTM networks (Du et al., 2015), it learns visual and temporal features but requires significant resources for design optimization. Contrary to $LSTM$-based methods, TeCNo (Czempiel et al., 2020) employs full temporal resolution and large receptive fields for surgical phase prediction. It leverages causal, dilated convolutions for online, fast inference on entire video sequences. Yi *et al.* (Yi et al., 2022) explore various multi-stage architectures by combining pre-trained models for solving surgical phase recognition tasks. Trans-SVNet (Gao et al., 2021) utilizes transformer architecture to fuse different embeddings for better capturing spatiotemporal information. Recent works (Basu et al., 2024; Fujii et al., 2024; Jamal and Mohareri, 2023) enhance MAE with improved masking procedures, such as estimating high-information regions, deriving masks from gaze-capturing data, or sampling tokens from high-information spatiotemporal areas instead of using random masking.

## 3. Methodology

Our proposed VideoMAE-based architecture has two parts: the autoencoder, reconstructing input video frames, and the downstream pathway classifying surgical events. They can be trained jointly or sequentially. The overall pipeline is illustrated in Figure 1. First, we outline our SurgFlowMAE methodology and introduce our novel masking strategy. Then, we describe the multitask approach and conclude with our objective functions.

### 3.1. Definitions

We are given a dataset of video sequences $V = \{X_1, ..., X_T\}$, and event labels $Y = \{y_1, ..., y_T\}$, where $T$ is the total number of frames in $V$, and $X_t \in \mathcal{R}^{H \times W \times C}$ is the video frame at time step $t$ with $H, W, C$ defining the frame's width, height and the number of channels. We estimate the magnitude of the optical flow $F_t \in \mathcal{R}^{H \times W}$ for each frame $X_t$. From now on, we will refer to it as optical flow. Our SurgFlowMAE model, defined by $f_\theta$ and parameterized by $\theta$, receives pairs of video frames and optical flow and outputs the reconstructed video together with the predicted class label.

$$V', Y' = f_\theta(V, F) \tag{1}$$

**Optical Flow Estimation**   The optical flow is precomputed using the SEA-RAFT algorithm (Wang et al., 2025) for each frame, providing a robust measure of motion and activity. We calculate it by analyzing frame differences over a temporal window of 1 second. Optical flow magnitude $F$ is defined as $F = \sqrt{u^2 + v^2}$, where $u$ and $v$ represent the horizontal and vertical components, respectively.

### 3.2. Video Masked Autoencoder

**Mask Sampling**   In conventional Masked Autoencoders, mask generation typically involves random patch selection. We hypothesize that regions with high motion, hence, large optical flow, carry important information for recognizing surgical actions. Thus, we propose a new sampling strategy (*cf.* Figure 2) based on the estimated prior probability distribution $P$. The choice of retaining or removing the regions according to the probability map from the frame $X_t$ depends on the training strategy, sequential or multitask.

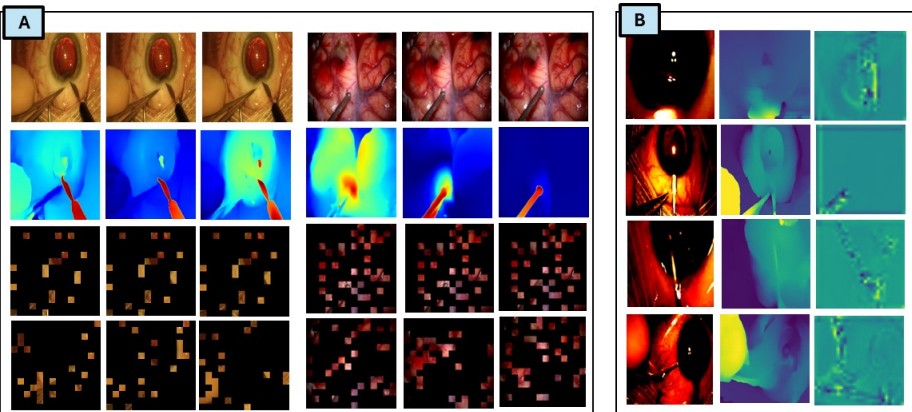

Figure 2: Visualization of Sampling Strategies: Subfigure (A) provides an overview of sampling techniques, displaying (from top to bottom) RGB images, optical flow representations, random tube masking, and flow masking. The left side shows CATARACTS, while the right side presents Neurosurgical data. Subfigure (B) illustrates the encoder's impact on feature representation, demonstrating examples where significant features do not always correspond to areas of highest motion. RGB images, optical flow, and encoded features arranged in columns.

Each frame $X_t$ is divided into a set of non-intersected patches $B$. The sampling process involves drawing $k$ patches from $B$ according to the probability distribution $P$, $B_k \sim P$  . We compare two approaches for calculating probability distribution. The first focuses on patches with higher motion dynamics. To increase reconstruction task complexity, we encourage the network to attend to masked regions, enhancing the pretrained model's effectiveness. We use min-max normalization $\|F\| = \frac{F}{F_{max}+\epsilon}$ to bring optical flow values into the $(0,1)$ range. The probability of selecting a patch is inversely proportional to optical flow magnitude $P = 1 - \|F_B\|$. In the multi-task training strategy, we reverse masking to keep the most informative parts visible, tailoring the reconstruction task to better align with the specific downstream task, with probability distribution proportional to optical flow magnitude $P = \|F_B\|$. We observe from surgical videos that the surgeon's hands sometimes exhibit more gestures compared to the tools in the scene. However, hand movement does not contribute to understanding the current phase. Thus, higher flow dynamics might be a sub-optimal feature for classifying the event. To address this concern, we extend the model $f_\theta$ (Equation 1) with an additional encoder $e$ that estimates the probability distribution $P$ for a set of patches $B$ from optical flow $F_B$.

$$V', Y' = f_\theta(V, e(F)) \tag{2}$$

In order to facilitate the gradient flow, we concatenate the encoded optical flow to the input frames $[e(F_t), X_t]$ as an additional channel. Hence, the probability map is implicitly learned throughout the optimization of the model $f_\theta$.

**Multitask Model** The sequential approach is extended using a multitask training strategy, where tasks are learned jointly. This model has two distinct output heads: the reconstruction head operates as a decoder, and the classification head consists of linear layers (Figure 1-A and Figure 1-B). The reverse masking choice described previously is validated by comparisons in Table 6, visualized in Figure 4. The encoder-decoder path for reconstruction follows the VideoMAE framework (Tong et al., 2022). For classification, encoded features pass through a pooling layer before entering a classification head with two fully connected layers: the first has 256 hidden dimensions with ReLU activation and dropout, and the final layer produces class logits.

### 3.3. Objective Functions

In a multitask learning strategy, the overall loss function is a weighted combination of two task-specific losses, while the two-step training strategy employs each loss independently.

$L_{\text{total}} = \alpha \cdot L_{\text{rec}} + \gamma \cdot L_{\text{CE}}$, where $\alpha$ and $\gamma$ are the weighting terms for the reconstruction and classification losses, respectively. Mean Squared Error (MSE) is used for the reconstruction task, measuring the difference between pixel values of original and reconstructed frames. The classification loss is calculated using Cross Entropy Loss, which evaluates the model's output probability between 0 and 1.

## 4. Experiments & Results

The foundational architecture of our model is derived from VideoMAE (Tong et al., 2022), specifically employing the ViT-Small backbone. We conduct experiments with two distinct variations of this model: the first is pretrained on the Kinetics-400 Action dataset, followed by fine-tuning on our specific use case dataset. The second variation involves training the reconstruction model from scratch.

### 4.1. Datasets

In our experiments, we use three medical datasets: CATARCTS (Al Hajj et al., 2019) for fine-grained surgical activity recognition, an in-house Neurosurgical dataset focusing on bleeding-related adverse events, and Ego-Surgery dataset (Fujii et al., 2024) for general phases in egocentric open surgery.

**CATARACTS** The dataset contains 50 cataract surgery videos, each at $1920 \times 1080$ pixels and 30 fps, annotated with 19 surgical phases. It is split into 25 training, 5 validation, and 20 testing videos, consistent with prior work (Köksal et al., 2024) for fair comparison.

**Microscopic Neurosurgery** The dataset for this study consists of 12 neurosurgical videos recorded at a resolution of 1920 x 1080 pixels and a frame rate of 60 fps. Annotations focus on two classes: "adverse bleeding event" and "non-adverse event". Due to the nature of the surgical scene, where bleeding is common—particularly from the opening of the dura mater—such instances are not classified as adverse events. Adverse events are specifically annotated when unintentional damage occurs due to tool-tissue interaction, necessitating immediate surgical intervention. This task is more complex than merely distinguishing between bleeding and non-bleeding scenarios. In our dataset, there are 205 occurrences of

Table 1: Comparison to SOTA on Cataracts. *: Own implementation, †, ‡: pretrained on K400, K400+Cataracts, respectively. St., Dyn.: Static and Dynamic Graph.

| Method | Training Strategy | End2End | Acc1 | Acc5 | Prec. | Rec. | Jacc. | F1 |
|---|---|---|---|---|---|---|---|---|
| DINO-TCN++ (Köksal et al., 2024) | Supervised | No | 77.0 | - | - | - | - | 74.4 |
| Xception-TCN++* | Supervised | ✓ | 75.2 | - | - | - | - | 74.4 |
| Holm (St.) (Holm et al., 2023) | Semi-supervised | No | 64.3 | - | - | - | - | 50.0 |
| Holm (Dyn.) | Semi-supervised | No | 75.2 | - | - | - | - | 68.6 |
| SANGRIA (Dyn.) (Köksal et al., 2024) | Unsupervised | ✓ | 83.4 | - | - | - | - | 78.2 |
| VideoMAE (Tong et al., 2022)*† | Reconstruction | No | 76.3 | 96.1 | 48.5 | 53.9 | 47.9 | - |
| ‡ | Reconstruction | No | 75.8 | 93.9 | 48.7 | 52.5 | 48.0 | - |
| SurgFlowMAE (Ours) ‡ | Multitask | ✓ | **87.5** | 97.1 | 77.9 | 80.6 | 76.6 | - |

Table 2: Ablation study on masking strategies for Flow Mask Models on Cataracts pretrained on K400

| Mask | Rec + Cls | | | | | Multitask | | | | |
|---|---|---|---|---|---|---|---|---|---|---|
| | Acc1 | Acc5 | Prec. | Rec. | Jacc. | Acc1 | Acc5 | Prec. | Rec. | Jacc. |
| Random | 75.8 | 93.9 | 48.7 | 52.5 | 48.0 | 85.4 | 96.0 | 75.8 | 77.6 | 74.9 |
| Encoder | 81.0 | **97.1** | 59.4 | 62.2 | 58.6 | 86.4 | 97.1 | 75.5 | 78.8 | 74.2 |
| Flow | **81.7** | 96.8 | **61.1** | **64.6** | **60.2** | **87.5** | **97.1** | **77.9** | **80.6** | **76.6** |

adverse events, while the remaining 1,673 sequences are categorized as normal events. The dataset is divided into training, validation, and test sets, comprising 70%, 15%, and 15% of the total data, respectively. A patient-wise split is implemented to prevent data leakage, ensuring that no patient appears in more than one subset, thereby enhancing the model's generalizability.

### 4.2. Experimental Setup

**Implementation Details** We utilize the ViT-Small backbone with an input patch size of $(16, 16)$ for all models. The input video and optical flow are processed at a resolution of $224 \times 224$ pixels, comprising 16 frames with a sampling rate of 2. For pre-training, in accordance with best practices established in prior research, the sampling ratio of input tokens is fixed at 90%. Additional implementation details can be found in section A.

**Evaluation Metrics** We assess our method using five benchmark metrics for surgical phase recognition and event classification: Accuracy (Acc1), Top-5 Accuracy (Acc5), Precision, Recall, and Jaccard index. We report micro-average accuracy for CATARACTS to compare with SOTA, while macro-average is used for other metrics to ensure equal importance across smaller classes.

### 4.3. Quantitative and Qualitative Results

**CATARACTS** In Table 1, we compare recent methods and SOTA approaches for phase segmentation on the CATARACTS dataset. The evaluated methodologies include long-range temporal learning techniques using DINO-TCN++ (Köksal et al., 2024), which fea-

Table 3: Cross-validation Results for Neuro Multitask Training from Scratch

| | Random | | | | Flow | | | | Encoder | | | |
|---|---|---|---|---|---|---|---|---|---|---|---|---|
| | Acc1 | Prec. | Rec. | Jacc. | Acc1 | Prec. | Rec. | Jacc. | Acc1 | Prec. | Rec. | Jacc. |
| **CV 1** | 70.3 | 71.2 | 70.3 | 67.1 | 76.8 | 74.8 | 76.8 | 73.2 | **89.5** | 80.1 | 81.1 | 77.8 |
| **CV 2** | 77.9 | 73.4 | 77.9 | 77.2 | 79.8 | 74.1 | 79.8 | 73.4 | 79.4 | 74.3 | 79.6 | 73.9 |
| **CV 3** | 77.6 | 77.8 | 77.5 | 74.7 | 84.4 | **83.3** | **84.4** | **81.5** | 83.9 | 81.0 | 83.4 | 79.6 |

tures a two-step model with DINO as a feature extractor followed by temporal classification. This is compared to our end-to-end variation that jointly trains Xception and TCN++. Graph-based approaches from (Holm et al., 2023) and (Köksal et al., 2024) examine static versus dynamic scene graphs. We also assess the masked encoder approach from (Tong et al., 2022), which serves as the foundation for our work. Our results show that the proposed SurgFlowMAE achieves competitive performance, with the multitask model outperforming the SOTA by 5% (4 points) in accuracy.

In Table 2, we analyze results from various model configurations, distinguishing between two-step models that separately learn reconstruction and classification tasks and the multitask model. Here, ' Mask' denotes different training strategies used for training and testing the models. *Flow* refers to masking simply based on the flow map input (Equation 1), while *Encoder* is the extended version, where masking is performed based on the estimated probability of the input flow maps (Equation 2). We assess the impact of masking types on both models trained from scratch and those fine-tuned from pretrained versions. Our analysis reveals three key insights. First, fine-tuning pretrained models from different domains (K400) benefits both setups. Second, flow-based masking consistently outperforms random tube masking, which selects the same patches for all frames in a sequence. This results in about 8% (5.90 points) improvement in accuracy (Acc1) for the K400+Cataracts configuration in the Rec+Cls task and a 2.5% (2 points) enhancement in multitask mode. Third, the multitask approach consistently surpasses two-step models, showing that learning reconstruction aids phase recognition. Extended results are shown in Table 5.

**Microscopic Neurosurgery**    Identifying adverse events in neurosurgery, where only surgical videos are available, poses significant challenges, as detailed in section 4.1. In Figure 5, we illustrate that not all bleeding instances are damaging events. While previous research has focused on bleeding detection in endonasal surgery (Pangal et al., 2022) and neurosurgical craniotomy (Tang et al., 2022), we are the first to introduce adverse recognition in microscopic neurosurgical videos. We utilize the multitask strategy for our analysis, which has proven most effective in the CATARACTS study presented in Table 2. Given that our dataset is approximately 2.5 times larger in duration, we choose to train the model from scratch. This decision is supported by the ablation study shown in Table 7. Damage events are closely linked to tool-tissue interactions, as hypothesized. This is supported by substantial improvements in flow masking, which targets regions of interest. In Table 3, we present results from patient-wise cross-validation using three non-intersecting splits. All three splits achieved an accuracy above 79.4%, with the best split reaching 89.5%. Across all splits, the optical flow and encoder-based masking methods consistently outperformed the baseline, a trend that is also observed in the experiments on the Egosurgery dataset,

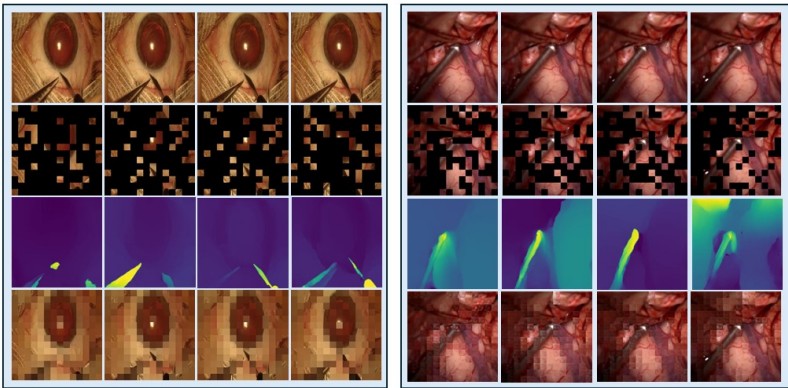

Figure 3: Visualization of the reconstruction output from the multitask flow model, organized into four rows: input RGB, masked RGB, optical flow input, and reconstructed output. Examples are shown for both CATARACTS and Neurosurgery.

Table 4: Inference Speed Analysis for Multitask Models

| Random | Flow | Encoding |
|--------|------|----------|
| 13 ms | 13.6 ms | 40 ms |

as reported in Table 8. We further conduct a qualitative assessment of the reconstruction output from the multitask model, as shown in Figure 3. The results indicate that masking 50% of the image while retaining key regions based on optical flow enhances the model's ability to reconstruct contextual information and the interactions between tissue and the surgical tool.

**Inference Analysis** Table 4 shows the runtimes for each 16-frame video sequence. While we are not targeting real-time applications, our neurosurgical videos are downsampled to 5 fps (200 ms between frames), making runtimes of 13-40 ms suitable for real-time scenarios, provided hardware requirements are met.

## 5. Conclusion

Understanding surgical workflows is essential for optimizing medical procedures. Distinct events, including surgical phases and tool-tissue interactions, provide critical insights. We propose a novel workflow recognition method that leverages optical flow information to enhance the capabilities of the baseline established by VideoMAE. Our approach focuses particularly on significant regions within the surgical scene. Our experiments, conducted across two diverse datasets, demonstrate that our approach considerably improves event classification accuracy. Specifically, for the CATARACTS dataset, we achieve a notable improvement of 5% over the state-of-the-art and 27% improvement on the Microscopic Neurosurgery dataset. Looking ahead, we aim to conduct a deeper investigation into adverse event recognition using a broader range of clinical data, paving the way for the anticipation of critical surgical events and further advancements in surgical workflow optimization.

## Acknowledgments

This work was conducted at and financed by the Corporate Research and Technology department of Carl Zeiss AG

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

## Appendix A. Implementation Details

**Implementation Details**   We utilize the ViT-Small backbone with an input patch size of $(16, 16)$ for all models. The input video and optical flow are processed at a resolution of $224 \times 224$ pixels, comprising 16 frames with a sampling rate of 2. For pre-training, in accordance with best practices established in prior research, the sampling ratio of input tokens is fixed at 90%. We employ the AdamW optimizer, configured with a weight decay of $1e - 4$ and betas set to $(0.9, 0.999)$. The pretraining phase utilizes a batch size of 32 and is conducted over 1200 epochs. As for the finetuning models, the classification head is finetuned for 350 epochs with cross-entropy and a batch size of 12.

## Appendix B. Supporting Results

### B.1.   CATARACTS

Table 5: Summary of Experiments and Results for Flow Mask Models

| Experiment & Pretrain Model | Masking Type | Metrics | | | | |
|---|---|---|---|---|---|---|
| | | **Acc1** | **Acc5** | **Precision** | **Recall** | **Jaccard Index** |
| **Rec + Cls** | | | | | | |
| K400 | Random | 76.26 | 96.13 | 48.52 | 53.94 | 47.87 |
| K400 + Cataracts | Random | 75.76 | 93.94 | 48.65 | 52.48 | 48.04 |
| K400 + Cataracts | Flow | **81.65** | 96.80 | **61.13** | **64.58** | **60.18** |
| K400 + Cataracts | Encoder | 80.98 | **97.14** | 59.40 | 62.17 | 58.57 |
| Cataracts | Random | **56.40** | **86.53** | 16.77 | 23.03 | 16.71 |
| Cataracts | Flow | 55.22 | 84.34 | 14.25 | 21.36 | 14.22 |
| Cataracts | Encoder | **56.40** | 85.19 | **17.16** | **23.55** | **17.10** |
| **Multitask** | | | | | | |
| Cataracts | Random | 61.95 | 90.75 | 39.71 | 44.95 | 38.55 |
| Cataracts | Flow | **66.67** | **91.75** | **41.14** | 45.16 | **39.54** |
| Cataracts | Encoder | 63.13 | 86.36 | 40.74 | **46.27** | 39.26 |
| K400 + Cataracts | Random | 85.35 | 95.96 | 75.79 | 77.62 | 74.89 |
| K400 + Cataracts | Flow | **87.54** | **97.14** | **77.87** | **80.63** | **76.62** |
| K400 + Cataracts | Encoder | 86.7 | **96.97** | 76.54 | 79.00 | 75.13 |

Table 6: Comparison of **Multi** Models with Different Masking Types

| Model | Mask Type | CATARACTS | | | | | K400 + CATARACTS | | | | |
|---|---|---|---|---|---|---|---|---|---|---|---|
| | | Acc1 | Acc5 | Prec. | Rec. | Jacc. | Acc1 | Acc5 | Prec. | Rec. | Jacc. |
| **In** | 0.1 | 65.8 | 90.6 | 39.5 | 44.0 | 38.0 | 86.2 | 97.1 | 75.4 | 77.5 | 73.5 |
| | 0.9 | 65.1 | 89.7 | 38.2 | 42.4 | 36.8 | 87.0 | 96.1 | 75.6 | 78.4 | 74.6 |
| **Out** | 0.1 | 67.3 | 89.4 | 41.9 | 46.7 | 39.9 | 86.4 | 96.3 | 76.1 | 78.4 | 74.4 |
| | 0.9 | 63.5 | 87.2 | 32.8 | 36.9 | 31.5 | 82.0 | 96.0 | 65.6 | 70.1 | 64.5 |

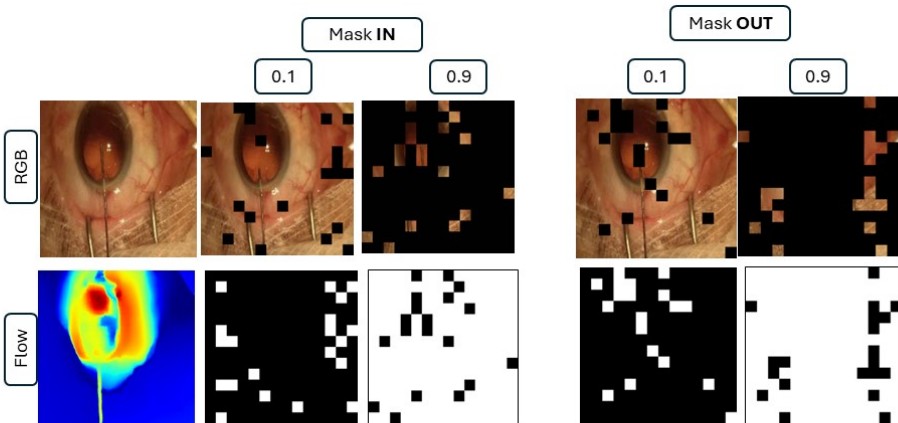

Figure 4: Example images illustrating different masking strategies and ratios. The first column shows the RGB image, followed by the corresponding optical flow. The subsequent columns depict the effects of masking strategies: "Masking In" at ratios of 0.1 and 0.9, which retain the informative regions, and "Masking Out" at ratios of 0.1 and 0.9, which remove the informative regions. These visualizations help to understand the impact of different masking techniques on the model's performance.

## B.2. Neuro

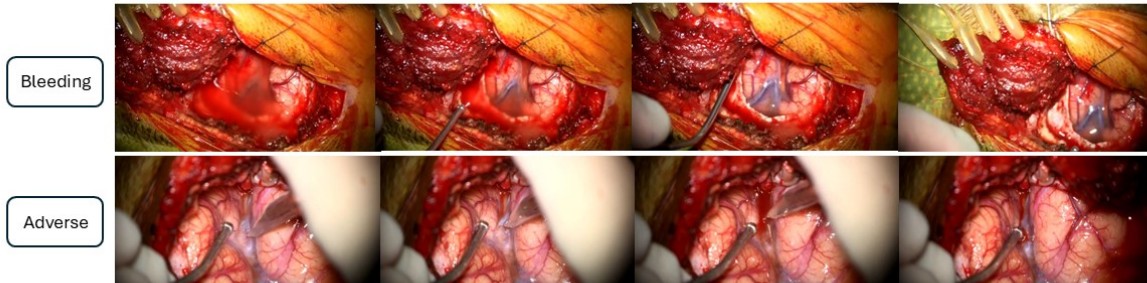

Figure 5: Comparison of surgical events in Microscopic Neurosurgery: The top row shows a non-adverse bleeding event, which is a common occurrence during surgery and does not indicate damage. The bottom row illustrates an adverse event caused by unintentional damage from surgical tools interacting with the tissue.

Table 7: Summary of Multitask Results for Flow Mask Models

| Exp. & Pretrain Model | Masking Type | Multitask | | | |
|---|---|---|---|---|---|
| | | **Acc1** | **Prec.** | **Rec.** | **Jacc.** |
| Neuro | Random | 70.3 | 71.2 | 70.3 | 67.1 |
| | Flow | 76.8 | 74.8 | 76.8 | 73.2 |
| | Encoder | 89.5 | 80.1 | 81.1 | 77.8 |
| K400 + Neuro | Random | 56.2 | 62.7 | 56.2 | 51.0 |
| | Flow | 66.8 | 68.4 | 66.8 | 62.7 |
| | Encoder | 68.8 | 64.4 | 57.8 | 53.6 |

## B.3. EgoSurgery

Table 8 shows the results of our experiments on phase recognition on the EgoSurgery dataset (Fujii et al., 2024) with different masking strategies. The results show that *Flow*-based masking achieves the best overall results in all metrics compared to other strategies.

Table 8: Phase Recognition Results for Egosurgery Dataset

| Mask | Accuracy (%) | Precision (%) | Recall (%) | Jaccard (%) |
|---|---|---|---|---|
| Random | 31.7 | 29.0 | 31.67 | 25.71 |
| Flow | **39.52** | **41.75** | **39.52** | **33.97** |
| Encoder | 38.57 | 31.43 | 38.57 | 29.05 |

