# OpenReview forum: "Surgical Flow Masked Autoencoder for Event Recognition"
_MIDL.io/2025/Conference — MIDL 2025 Poster_

### Official Review · Reviewer_GRdU · 2025-02-15

**Confidence:** 5
**Preliminary Rating:** 3
**Final Rating:** 3

**Summary:**

The authors propose to utilize the video masked autoencoder with their proposed mask sampling technique based on an estimated prior probability map derived from optical flow for surgical phase recognition and surgical event recognition. They experiment on two datasets and evaluate different training strategies to identify the best-performing model.

**Strengths:**

(1) The literature review explores a variety of techniques employed in surgical phase and workflow recognition. These range from frame-based modeling and small-resolution video temporal modeling to comprehensive full-resolution video temporal modeling.

(2) The authors conduct several ablation studies to select the best-performing model using different training strategies.

**Weaknesses:**

(1) Lack of comparison with state-of-art methods developed by others on open-source datasets.

(2) As the dataset size is small and highly unbalanced, cross-validation is needed for the Microscopic Neurosurgery dataset (12 videos, ~11% of sequences are surgical events).

(3) If the method is for real-time application, inference speed analysis is needed.

(4) For multi-task learning, it would be great to know the performance of the reconstruction task.

**Detailed Comments:**

For weakness point (1), while the authors selected several methods for comparison, all methods are their own implementations.

The results of the two methods were derived from their own implementation.

Similarly, two methods from K¨oksal et al. (2024) and Holm et al. (2023) also stem from their own implementation, due to overlapping author teams. K¨oksal et al. (2024) is a preprint.

I strongly recommend that the author utilize a more widely adopted open-source dataset like Cholec80 for their experiment for the following reasons. (For fine-grained surgical activity recognition, CholecT50 is a better choice. )

(a) Surgical phase/workflow recognition is a well-explored area with numerous prior studies. In fact, many papers in the related work section of this study have utilized the Cholec80 dataset for evaluation.

(b) The authors used two small datasets to evaluate their method. The Cholec80 dataset is a good fit for the author to conduct their experiment.

**Justification Of The Final Rating:**

I would like to express my appreciation to the authors for their response. However, the absence of a fair comparison with other state-of-the-art methods makes it challenging to evaluate the merits of their proposed approach. As a result, I cannot support the acceptance of this paper. That said, I acknowledge the authors' efforts in addressing other weaknesses highlighted during the review process, and therefore, I see no reason to lower my rating. My final evaluation remains unchanged at Borderline.

**Justification Of The Preliminary Rating:**

Considering the lack of evaluation and comparison with the state-of-the-art methods, I have to vote for Boardline. I encourage the authors to revise the paper, especially for the part of comparison with state-of-the-art methods developed by others on open-source datasets.

**Questions To Address In The Rebuttal:**

Please address weakness points (1) - (3). Especially weakness (1) and (2).

Weakness point (4) is optional to address.

**Special Issue:**

No

---

> ### Author Response · Authors · 2025-03-08
>
> We thank the reviewer for the insightfull feedback and detailed comments.
>
> - Additional experiments on public datasets:
>   -  We appreciate the reviewer for highlighting the Cholec dataset, and we would be happy to incorporate results from that dataset. However, due to time constraints and the extensive data processing and training efforts required, we believe it is not feasible to provide comprehensive experimental results on this dataset at this time. We would be glad to include these experiments in the camera-ready version of the manuscript.
>
>   - To further validate our work, we have conducted an experiment on the Egosurgery dataset, which is more closely aligned with our focus on open surgery. We selected 3 videos from this dataset for training, ensuring that they encompass all 9 annotated classes. We then tested the performance of our models on a fourth video that was not included in the training set. All four videos provide a similar view of patients during procedures, captured from an egocentric camera positioned on the surgeon's head, which offers a perspective relevant to our focus in neurosurgery. Our evaluation indicates that both of our variants—optical flow-based masking and the encoder—demonstrate superior performance compared to the benchmark VideoMAE. The results for EgoSurgery are reported in Table 8 in the paper.
>
>   - Cross-validation: We appreciate the suggestion regarding cross-validation on the neurosurgical dataset, and we addressed this by conducting experiments with three variants of multi-task models using three non-intersecting train/test splits. In each split, we utilized 8 videos for training and validation and 2 videos for testing. The results of these experiments are illustrated in Table 3, where we demonstrate that all three splits achieve an accuracy above 79.4%, with the best split reaching an accuracy of 89.5%. Across all splits, we observe that the optical flow and encoder-based masking methods outperform the baseline using the random masking approach described in VideoMAE.
>
>   - Runtime: We have added Table 4 with runtimes measured for each video sequence of 16 frames. While we do not aim for real-time application at this stage, our neurosurgical videos are downsampled to 5 fps, equating to 200 ms between consecutive frames. Thus, runtimes of 13-40 ms would be suitable for a real-time scenario, provided that hardware requirements and constraints are addressed.
>
>   - Reconstruction: We conducted a qualitative assessment of the reconstruction output from the multitask model, as shown in Figure 3. The results indicate that masking 50% of the image while retaining key regions based on optical flow enhances the model's ability to reconstruct contextual information and the interactions between tissue and the surgical tool.

---

> > ### Comment · Reviewer_GRdU · 2025-03-13
> > **Lack of comparison with state-of-the-art methods developed by others on open-source datasets**
> >
> > I appreciate the authors' responses, but the most significant weakness remains unaddressed: the lack of comparison with state-of-the-art methods developed by others on open-source datasets.

---

> > > ### Author Response · Authors · 2025-03-13
> > >
> > > We thank the reviewer for their valuable feedback and the suggestion to include additional comparisons across datasets. We will incorporate comparisons on the Cholec80 dataset in the camera-ready version, although we could not complete these experiments during the rebuttal period.
> > >
> > > Our choice of the CATARACTS dataset (publicly available) and our proprietary Neuro dataset was based on our focus on neuro and ophthalmic surgical procedures, intentionally excluding endoscopic surgeries. Due to the lack of publicly available neurosurgery datasets, we evaluated our methods and baseline algorithms using our own data. The baseline algorithms were selected from studies specifically addressing neuro or eye surgeries, including SANGRIA, rather than due to any author overlap.
> > >
> > > To further support our claims, we reviewed literature on endoscopic surgery and identified studies that also utilized the CATARACTS dataset. Below is a comparison of our proposed method with two additional approaches (Trans-SVNET (Gao et al., 2021) and SR-Mamba (Cao et al., 2024)  ) reporting their performance on this dataset:
> > >
> > > | Method                     | F1 Score |
> > > |---------------------------|----------|
> > > | Trans-SVNET (Gao et al., 2021) | 84.02    |
> > > | SR-Mamba (Cao et al., 2024)    | 85.38    |
> > > | **SurgFlowMAE (ours)**      | **86.5**     |
> > > | SANGRIA                   | 78.2     |
> > > | Holm (Dyn.)               | 68.6     |
> > >
> > >
> > > Since the existing methods primarily report the F1 score, we have computed this metric for our best-performing model as well. Additionally, we have included the F1 scores for our baseline methods to provide a comprehensive comparison.
> > > The results indicate that our approach outperforms all state-of-the-art methods on the CATARACTS dataset. We would be happy to incorporate this comparison into the camera-ready version of our manuscript.
> > > We appreciate the reviewer's understanding and look forward to enhancing our manuscript with the suggested comparisons.

---

> > > > ### Comment · Reviewer_GRdU · 2025-03-14
> > > > **Pre-training dataset is still different, fair comparison needed**
> > > >
> > > > I would like to thank the authors for their response. However, the pre-training methods remain distinct. While Trans-SVNET (Gao et al., 2021) and SR-Mamba (Cao et al., 2024) utilized ImageNet for pre-training, SurgFlowMAE was pre-trained using K400. Given that SurgFlowMAE/VideoMAE employed ViT as the backbone, could you please provide the ImageNet encoder pre-training results without the use of the K400 dataset to ensure a fair comparison?

---

> > > > > ### Author Response · Authors · 2025-03-15
> > > > >
> > > > > We would like to thank the reviewer for their prompt response and for reviewing the additional information we provided.
> > > > >
> > > > >
> > > > >
> > > > > We agree that more comparisons across additional datasets and methods are always beneficial and contribute to making the work more robust. In our study, we compare our method with six state-of-the-art approaches (four in the main paper and two additional methods introduced during the rebuttal), demonstrating the superior performance of our approach. We believe this is a sufficient amount of comparison and is comparable to other works in the community.
> > > > >
> > > > >
> > > > >
> > > > > Additionally, we conduct an extensive ablative study of our training strategy across two datasets (public and private).
> > > > >
> > > > >
> > > > >
> > > > > Our approach is grounded in prior art (VideoMAE), which offers pretrained weights on the K400, Something-Something V2, and UCF101 datasets. While the authors claim to utilize ImageNet-1K for initial pretraining, they do not provide a model that is solely trained on the ImageNet dataset.
> > > > >
> > > > > Using ImageNet features from official (facebookresearch/mae) requires changing the encoder layers and patch embedding dimensions to initialize the VideoMae architecture, which is suboptimal since the temporal dimension is underutilized. Therefore, relying solely on ImageNet features may disadvantage video-based architectures and we do not have the capacity to conduct complete retraining given the limited time frame.
> > > > >
> > > > >
> > > > >
> > > > > To further support our work, we conducted an additional experiment using a pretrained model on the Something-Something V2 dataset. The results are available below:
> > > > >
> > > > >
> > > > >
> > > > > | **Method**                             | **F1 Score** |
> > > > > |----------------------------------------|--------------|
> > > > > | Trans-SVNET (Gao et al., 2021)        | 84.02        |
> > > > > | SR-Mamba (Cao et al., 2024)           | 85.38        |
> > > > > | SANGRIA                                | 78.2         |
> > > > > | Holm (Dyn.)                            | 68.6         |
> > > > > | **SurgFlowMAE (ours) Pretrained from K400** | **87.5**     |
> > > > > | **SurgFlowMAE (ours) Pretrained from SSV2** | **87.2**     |
> > > > >
> > > > >
> > > > >
> > > > > For our evaluation, we followed the accepted procedures from public benchmarks and challenges such as the Critical Views of Surgery (CVS) Challenge, EndoVis, and FLARE, where authors present results for comparison without tuning their work to better fit the methods of competitors, which is deemed fair by the research community.

---

> > > > > > ### Comment · Reviewer_GRdU · 2025-03-17
> > > > > > **Follow up: fair comparison needed**
> > > > > >
> > > > > > In their latest response, dated March 15, the claims from the authors are unjustified.
> > > > > >
> > > > > > They assert that they compared "six state-of-the-art approaches." However, as I highlighted in my review, four of these are their own implementations (some are their prior work). Furthermore, in the two additional works included in the revision, the comparisons are unfair because different pre-trained weights were used. Specifically, Trans-SVNET (Gao et al., 2021) and SR-Mamba (Cao et al., 2024) leveraged ImageNet for pre-training, while SurgFlowMAE was pre-trained using K400.
> > > > > >
> > > > > > The authors argue that ImageNet weights are suboptimal due to the underutilization of the temporal dimension. However, this claim is inconsistent, as the temporal designs in Trans-SVNET and SR-Mamba were not pre-trained—only their ResNet feature extractors were. Additionally, the authors mention that prior work, such as VideoMAE, employs ImageNet-1K for initial pre-training. This, in fact, demonstrates that utilizing ImageNet-1K for pre-training is a viable strategy for their own approach as their work is based on VideoMAE (The authors used VideoMAE pre-trained weights in the paper).
> > > > > >
> > > > > > Lastly, the claim about adhering to public benchmarks and challenges lacks merit without fair comparisons to other works. While challenges permit flexibility, including the use of additional datasets as long as the rules are followed, publications demand fair and transparent comparisons to justify the design and validity of the proposed work.

---

### Official Review · Reviewer_zM9w · 2025-02-19

**Confidence:** 4
**Preliminary Rating:** 2

**Summary:**

The authors study the problem of classifying surgical event from videos. They propose to use video masked autoencoders (VideoMAE) as the additional training objective (in addition to the standard cross-entropy loss for classification). A novel masking strategy is proposed to assign different probabilities to patches with faster motion. Specifically, the patches with faster motion get a higher probability of being masked when training alone, and a lower probability when trained jointly with the segmentation task. The authors validate the method on a public dataset and a private one, where the proposed method surpassed VideoMAE and other methods.

**Strengths:**

The discovery of sampling masks based on optical flow is interesting as it leverages motion cues to refine mask selection during the MAE. Moreover, the use of multi-mask training, while a well-established technique in the field, gives major improvements on the classification accuracy.

**Weaknesses:**

1. The idea of sampling masks based on optical flow is interesting, but it means that the masks for different frames are different. This means the reconstruction task can be easier in the end since the network can rely on patches from adjacent frames. Can the authors provide more explanation on why this operation "increases the task complexity"? Furthermore, can authors compare to have different masks for each frame?
2. The comparison to other methods is a bit unclear in my option since (1) the proposed method utilized additional data by utilizing pre-trained weights (K400) while other methods except VideoMAE do not have this benefit; (2) a significant improvement is achieved by switching from sequential training to joint training, as demonstrated by Table 2. I think this is a simple strategy that could be shared by other methods, and if without this strategy, the proposed method is worse than SANGRIA on CATARACTS.

**Detailed Comments:**

The writing of the paper could also be improved. Specifically,
1. In Sec. 3.2, the authors first state that the sampling strategy is different for sequential or multitask, yet the introduction of the multitask is later introduced.
2. The "Training Strategy" in Table 1 is confusing. The author uses strategy to refer to network structure (graph), pre-trained dataset (K400), and training strategy (Multitask).
3. The "Mask" in Table 2/3/4 is not explicitly defined. For the "Flow" strategy, is encoder also utilized (Eq (2)) or not (Eq (1))?

**Justification Of The Preliminary Rating:**

While the work is interesting in discovering a new masking strategy, the improvement of this strategy alone is not very significant (2.1% acc improvement between random mask vs. the proposed method). And the setting for the competing methods could be further improved. Lastly, the clarity of the writing could also be improved.

**Questions To Address In The Rebuttal:**

1. Conduct an additional study of having different masking for each frame, as pointed out in Weaknesses 1.
2. Creating a more fair comparison by either granting access to pre-trained weights for other methods or imposing joint learning strategy to other methods.

---

> ### Author Response · Authors · 2025-03-08
>
> We thank the reviewer for their thoughtful review and for taking the time to evaluate our paper. We sincerely appreciate the insights and constructive feedback that helps us to further improve our work.
>
> - Writing:
>
>   - Table 1 has been adjusted following the reviewer's comments.
>
>   - We have added an explicit definition for masking strategies in Sec. 4.3. (Text highlighted)
>
> - Masking Strategies:
>
>   - We chose to implement tube random masking based on a key consideration: it has been demonstrated to be beneficial compared to fully random masking by the authors of VideoMAE. Thus, we compare only with the state-of-the-art approach that utilizes tube masking.
>
> - Complexity of Masks:
>
>   - The optical flow-based masking strategy ensures consistent masking in the temporal domain. In contrast to the random masking strategy, where some objects may be masked in one frame and visible in another—making the reconstruction task easier—optical flow-based sampling ensures that the object is masked throughout the entire video sequence, thus making reconstruction more challenging for the model.
>
>   - Consequently, our approach masks the informative regions where tool-tissue interactions occur, directing the network's focus to these areas. The model must learn to reconstruct the scene without relying on easily accessible information from high-motion areas, which challenges it to develop a deeper understanding of the context and relationships within the data. This strategy ultimately enhances the robustness of the model by encouraging it to learn from less obvious cues rather than relying on direct visual information.
>
> - Comparison with Other Methods:
>
>   - **Utilization of Pre-trained Weights and Joint Training Strategy**: We would like to clarify that the use of pre-trained weights is not applicable for graph-based approaches such as SANGRIA, as no pre-trained weights are available for this architecture. In our comparisons with VideoMAE, we present two scenarios: one utilizing a pre-trained model and another training from scratch. Additionally, Sangria is an end-to-end approach that is similar to the joint training strategy.

---

> > ### Comment · Reviewer_zM9w · 2025-03-14
> >
> > Thank you for your response. However, my main concerns remain:
> > 1. The authors state that "the object is masked throughout the entire video sequence," but this does not appear to be entirely accurate. In Figures 2 and 3, parts of the objects remain visible. Additionally, as described in the paper, the mask is sampled inversely proportional to flow magnitude rather than deterministically—masks are provided only when the flow magnitude exceeds a certain threshold.
> > 2. Consequently, my concern about the limited impact of this strategy remains. The reported improvement (2.1% accuracy when switching from random masking to the proposed method) is not particularly significant. Most of the observed performance gains seem to stem from leveraging a strong pre-trained dataset and the usage of multi-task learning.

---

> > > ### Author Response · Authors · 2025-03-14
> > >
> > > Dear Reviewer,
> > >
> > > Thank you for your response.
> > >
> > > 1. **Clarification of Masking Strategy**
> > >    We would like to clarify that our masking strategy is distinct from the two training strategies (Reconstruction plus Classification and Multi-task). As stated in Section 3.2 of the paper, in the multi-task training strategy:
> > >
> > >    > "We utilize reverse masking to keep the most informative parts visible, aligning the reconstruction task with the downstream task based on a probability distribution proportional to optical flow magnitude."
> > >
> > >    a. **Multitask Flow Model Output**
> > >    Figure 3 shows the reconstruction output from the multitask flow model, which consequently has parts of the objects (tools) visible due to the large optical flow magnitude in these areas; hence, we focus on the most informative parts by masking.
> > >
> > >    b. **2-Step Approach Output**
> > >    Figure 2 shows the reconstruction output from the 2-step approach. As depicted in the figure, regions with high optical flow are masked out according to the masking strategy, making the reconstruction model stronger. We provide in Figure 4 more masking examples and comparisons of different ratios.
> > >
> > >    Please note that in this case, we sample from the probability distribution; although high optical flow regions have a lower probability of being selected, they can still be sampled. This is a strength of our method, as it does not use a threshold, which encourages the network to attend to the important regions.
> > >
> > > 2. **Performance Improvement**
> > >    We refer to Table 2, where the improvement of our method (Classification + Reconstruction) shows a 6% accuracy increase compared to random masking. This indicates that the performance gains are not solely due to multi-task learning, as suggested. This improvement is already comparable to the SOTA for CATARACTS phase recognition, while our best model (multitask) beats the SOTA by 3%.

---

> ### Comment · Area_Chair_UPn3 · 2025-03-17
> **Final rating is missing**
>
> Dear zM9w,
>
> Would you be so kind to provide your final rating? Thanks!
>
> AC

---

> > ### Comment · Reviewer_zM9w · 2025-03-18
> >
> > Dear AC,
> >
> > My final rating remains the same as my initial rating (weak reject), which is why I did not update it. Sorry for the confusion.
> >
> > While the authors’ latest response clarifies the method details, my concerns about its effectiveness remain. The method gained superior performance by having strong pre-trained weights and utilizing multi-task learning, while the improvement from the new masking strategy is not significant in the optimal setting. Also, I agree with Reviewer GRdU that the comparisons are somewhat unfair and limited to the authors' prior work.

---

### Official Review · Reviewer_oqKb · 2025-02-26

**Confidence:** 3
**Preliminary Rating:** 4
**Recommendation:** Poster

**Summary:**

The paper proposes a mask sampling technique based on an estimated prior probability map derived from optical flow to enhance the performance of VideoMAE for surgical event recognition. The authors implemented the SEA-RAFT algorithm (Wang et al., 2025) to compute optical flow, which was then max-min normalized and treated as a probability distribution for identifying valuable regions. Two sampling methods were introduced: one selecting patches with higher motion dynamics, and the other extending the pretrained model with an additional classification module for recognizing hand movements. The authors used ViT-Small as the backbone model and evaluated their approach on two datasets. The proposed method achieved a 5% improvement on the CATARACTS dataset and a 27% improvement on the neurosurgical dataset.

**Strengths:**

The method is easy to adapt and shows impressive improvements of 5% and 27% on two datasets. Using optical flow for focused attention to focus on important regions in surgical videos is an interesting approach. Trying to classify the hand movements of  surgeons helps achieve better performance and also provides valuable insights into the field

**Weaknesses:**

The experimental results lack clarity and require a more detailed explanation of the setup. Relying on a single backbone model limits the reliability of the findings, as performance may vary with different architectures. Additionally, the paper’s presentation is a notable weakness, as it is difficult to read and follow.

**Detailed Comments:**

I suggest more experiments on other backbone models and improve the writing on the results part.

**Justification Of The Preliminary Rating:**

The proposed method heavily depends on the performance of the pretrained and backbone models, which raises some concerns. I also have doubts about the impact of mask sampling on the improvement, as the CATARACTS dataset is relatively small. However, the approach itself is valuable for advancing computer-assisted surgery.
Weak accept.

**Questions To Address In The Rebuttal:**

No question

---

> ### Author Response · Authors · 2025-03-08
>
> We sincerely thank the reviewer for recognizing the value of our work and for providing constructive feedback that will help strengthen our paper.
>
> - The motivation behind our flow-based mask sampling strategy stems from a key observation in surgical videos: regions with significant motion typically correspond to areas of high clinical importance. By directing the network's attention to these dynamic regions, we enhance its ability to capture relevant features for downstream tasks such as phase recognition. Our experimental results consistently validate this approach. As demonstrated in Tables 2, 5, and 8, flow-based masking significantly outperforms random masking across all evaluated scenarios.
>
> - We have  improved the clarity of the tables captions in the results section and will continue improving it further.

---

> > ### Comment · Reviewer_oqKb · 2025-03-14
> >
> > Thank you for your detailed response and clarifications. I appreciate the effort to improve the clarity of the results section. However, the experimental results still remain unconvincing because of insignificant improvement. That said, I value your insightful observation in surgical videos and believe it is useful in the medical domain. Therefore, I keep my decision as Weak Accept.

---

> ### Comment · Area_Chair_UPn3 · 2025-03-17
> **Final rating is missing**
>
> Dear oqKb,
>
> Would you be so kind to provide your final rating? Thanks!
>
> - AC

---

> > ### Comment · Reviewer_oqKb · 2025-03-17
> >
> > Dear AC,
> >
> > I don't know why I cannot edit my review (Edit button disappears) to add my final rating.
> >
> > Btw, my final rating is Weak Accept.

---

### Author Rebuttal · Authors · 2025-03-08

**Rebuttal:**

We highly appreciate the constructive feedback from the reviewers. We have revised the paper and respond to the concerns raised by the reviewers below.

**Supporting Material:**

/attachment/c5a3497131f0fe24e661f24f0e57e6a0d56ff002.pdf

---

### Comment · Area_Chair_UPn3 · 2025-03-08
**Time for discussion and review of the rebuttal**

Dear reviewers,

It is now time to consider the responses from the authors. If you are or are not satisfied with author's reply please still post to openreview your feedback to the rebuttal and update your scores. Especially, please update the scores if you feel that the authors have addressed your concerns.

Please note that you can and **are encouraged** to discuss the scores of other reviewers if you disagree with them to make the best

As AC, my responsibility is to post meta-reviews by March 21st, and I would thus like to kindly ask you to consider the authors' rebuttal as soon as possible.

// Your Area Chair

---

### Meta-Review · Area_Chair_UPn3 · 2025-03-24

**Recommendation:** Accept (Poster)
**Confidence:** 3

**Metareview:**

This paper has three reviews: Weak reject, weak accept, and borderline. The weak accept review by oqKb is rather shallow, and based on my analysis of other reviews by zM9w and GRdU (kudos for deep analysis), the paper needs to have a better clarity.

What I do not agree with from GRdU, is how authors should have approached the comparison. It is totally OK to do reimplementation. This, however, has nothing to do with this study and its merit, in my opinion.

All in all, after reading the reviews and the paper, I tend to consider that the paper is still in the accept category. The method proposed by the authors has merit, and has motivation.

/ Area Chair